# Conserved strategies of RNA polymerase I hibernation and activation

Florian B. Heiss [1], Julia L. Daiß[1], Philipp Becker[1] & Christoph Engel [1✉]

RNA polymerase (Pol) I transcribes the ribosomal RNA precursor in all eukaryotes. The mechanisms 'activation by cleft contraction' and 'hibernation by dimerization' are unique to the regulation of this enzyme, but structure-function analysis is limited to baker's yeast. To understand whether regulation by such strategies is specific to this model organism or conserved among species, we solve three cryo-EM structures of Pol I from *Schizosaccharomyces pombe* in different functional states. Comparative analysis of structural models derived from high-resolution reconstructions shows that activation is accomplished by a conserved contraction of the active center cleft. In contrast to current beliefs, we find that dimerization of the *S. pombe* polymerase is also possible. This dimerization is achieved independent of the 'connector' domain but relies on two previously undescribed interfaces. Our analyses highlight the divergent nature of Pol I transcription systems from their counterparts and suggest conservation of regulatory mechanisms among organisms.

---

[1] Regensburg Center for Biochemistry, University of Regensburg, Universitätsstraße 31, 93053 Regensburg, Germany. ✉email: christoph.engel@ur.de

Transcription of the ribosomal RNA (rRNA) precursor by RNA polymerase (Pol) I is a prerequisite for the biosynthesis of ribosomes in all known eukaryotes[1]. To allow transcription initiation in baker's yeast *Saccharomyces cerevisiae* (*Sc*), monomeric Pol I is bound by the initiation factor Rrn3, enabling recruitment to the rDNA promoter via the heterotrimeric core factor (CF), TBP and upstream activating factor (UAF)[2–4]. Structures solved from *Sc* Pol I crystals revealed inactive polymerase dimers with widely expanded active center clefts in three similar conformations[5–7], matching biochemical observations in extracts[8], initial and recent Electron Microscopy studies[9,10]. Such dimerization relies on a 'connector' domain at the C-terminus of Pol I subunit A43. In detail, the 'stalk' subunit complex of one monomer is inserted into the 'cleft' of the other monomer from the upstream side. This allows formation of a connector α-helix in subunit A43 which attaches to the clamp core helices of subunit A190 in the second monomer. Furthermore, a C-terminal connector β-hairpin traps the lid of subunit A190, completely inactivating both polymerases[5,6]. Activation then requires Pol I monomerization, cleft contraction and stabilization of monomers by Rrn3. Further cleft contraction takes place upon promoter melting and Pol I interaction with the DNA/RNA hybrid[11–13].

Structures of monomeric states and of actively elongating Pol I from *Sc* were solved by single particle Electron Cryo Microscopy (cryo-EM)[14,15]. Furthermore, differential fluorescent allele tagging demonstrated that Pol I dimerization is associated with inactivation (hibernation) under certain starvation conditions in an elegant in vivo approach[16]. In such experiments, deletion of the connector resulted in dimer disruption. To date, it is unclear which mechanisms and structural features of Pol I regulation are specific to *Sc* and which are conserved among organisms[17].

Here, we use single particle cryo-EM to solve the structure of Pol I from *Schizosaccharomyces pombe* (*Sp*) in a monomeric (apo) and an actively elongating form at high resolution, showing that Pol I cleft contraction upon transcription activation is common to both organisms. Strikingly, we also uncover that Pol I dimerization can take place independent of the A43 connector domain in vitro and solve the cryo-EM reconstruction of such a dimer. Our results allow discussing the evolutionary conservation of structural features, hibernation and activation mechanisms.

## Results

**Preparation and cryo-EM of *Schizosaccharomyces pombe* Pol I.** To study the structure of its Pol I, we generated an *Sp* strain carrying a C-terminal flag-his tag on subunit AC40. Purification using established protocols[5] yielded pure Pol I that shows protein bands for all 14 subunits (Supplementary Fig. 1). *Sp* Pol I is active in elongation and cleavage of an RNA primer from synthetic constructs in vitro similar to *Sc* Pol I (Supplementary Fig. 2). To determine the structure of a monomer, we stabilized Pol I by crosslinking with BS3, followed by quenching and size exclusion chromatography (Methods). In an independent experiment, we established an *Sp* Pol I elongation complex (EC) in vitro similar to its *Sc* counterpart[15]. Using three locked nucleic acid (LNA) bases at the 3′ end of the RNA primer reduced sample heterogeneity by preventing transcript cleavage via subunit A12.2 (Methods). Two cryo-EM datasets were collected on a Titan Krios Electron Microscope equipped with Falcon III direct electron detector: one from non-crosslinked, LNA-containing EC particles and one from BS3-crosslinked Pol I, both following size exclusion chromatography (Supplementary Table 1).

**Pol I architecture is conserved but sub-complex occupancy varies among organisms.** The cryo-EM densities reveal

secondary structure elements for the entire polymerase and side chain orientations in many regions (Fig. 1; Supplementary Figs. 1–3). The architecture of *Sc* and *Sp* Pol I is similar, despite poor overall sequence identity (Supplementary Fig. 4, Supplementary Table 2). Both, EC and monomeric Pol I, lack cryo-EM density for the A49/A34.5 heterodimer, indicating that the sub-complex is flexible or was lost during grid preparation (Fig. 1). The presence of the A49/A34.5 subunit-complex in Pol I preparations was confirmed by Coomassie−stained SDS-PAGE (Supplementary Fig. 1a) and mass-spectrometry analysis.

Differences between *Sc* and *Sp* Pol I include the lack of density for the Pol-I-specific helix α0 in subunit Rpb6 and insertions in the 'Dock' and 'Foot' domains of *Sp* subunit A190. Helix α0 forms only in *Sc* Pol I and was presumed to strengthen stalk attachment compared to Pol II[5,6]. A divergent region (residues 549–559) and an insertion (residues 591–600) in the 'external 2' domain of *Sp* subunit A135 are adjacent to the lobe-region responsible for tight association of the A49/34.5 sub-complex[18] and may be responsible for its reduced affinity (compare Supplementary Figs. 3b and 4). Poor stalk density is observed, indicating a high degree of flexibility, similar to human Pol II[19] and potentially linked to a lack of Rpb6 α0 formation. Nevertheless, fitting of a stalk homology model created from the crystal structure of the *Sc* A14/A43 complex[20] was possible. In addition, local resolution is reduced in the jaw regions and the tip of the clamp domain of subunit A190 in *Sp* Pol I EC and monomer reconstructions, also indicating a high degree of flexibility (Supplementary Figs. 1 and 2). The C-terminal domain of subunit A12.2 is mostly flexible in ECs and monomers, although low-resolution density in the *Sp* EC (Supplementary Fig. 3a) may indicate a position outside subunit A135 as observed in some *Sc* ECs[21].

***Sp* Pol I contains an expander element that is flexible in ECs.** In ECs, we observe density for DNA and RNA, allowing us to build and refine a model for 25 template bases, 11 non-template bases and 7 RNA bases (Fig. 1d). Interactions with Pol I, positioning in the cleft and relative orientation of the scaffold are similar to *Sc* structures[14,15,21]. We do, however, observe a density that we attribute to the 'expander' ('DNA-mimicking') element in Pol I monomers (Fig. 1b). This density is located between the 'protrusion' domain of subunit A135 and the 'clamp core' region in subunit A190 on the upstream region of the active center cleft and appears to prevent unspecific DNA binding. Due to flexible connections to the 'jaw' domain of subunit A190, we refrained from modeling the exact residues of the section in the *Sp* monomer. Location of presumed expander density in the cleft is comparable to that observed in *Sc* Pol I dimers (Supplementary Fig. 3d), while the element is flexible in *Sc* Pol I monomers.

***Sp* Pol I dimerizes in vitro independent of the 'connector' domain.** Surprisingly, we observed many dimer-particles in the monomer dataset (Supplementary Fig. 1). As described above, Pol I subunit A43 does not contain a connector element in *Sp* and therefore should be unable to form dimers, as inferred from connector deletions in *Sc*[16]. Previous studies described that *Sc* Pol I dimers and monomers are in equilibrium in solution. Buffer conditions[8] and concentration of highly purified Pol I[5] can shift this equilibrium in vitro. To test whether dimerization is possible in solution or may result from chemical crosslinking, we carried out analytical size exclusion chromatography (SEC) in different buffers and in-solution dynamic light scattering (DLS) at different concentrations of *Sp* Pol I. A shift to earlier SEC elution volumes indicates an increased particle size under high-salt conditions (Fig. 2a). Negative stain EM analysis of SEC peak fractions revealed the presence of dimeric particles independent of the

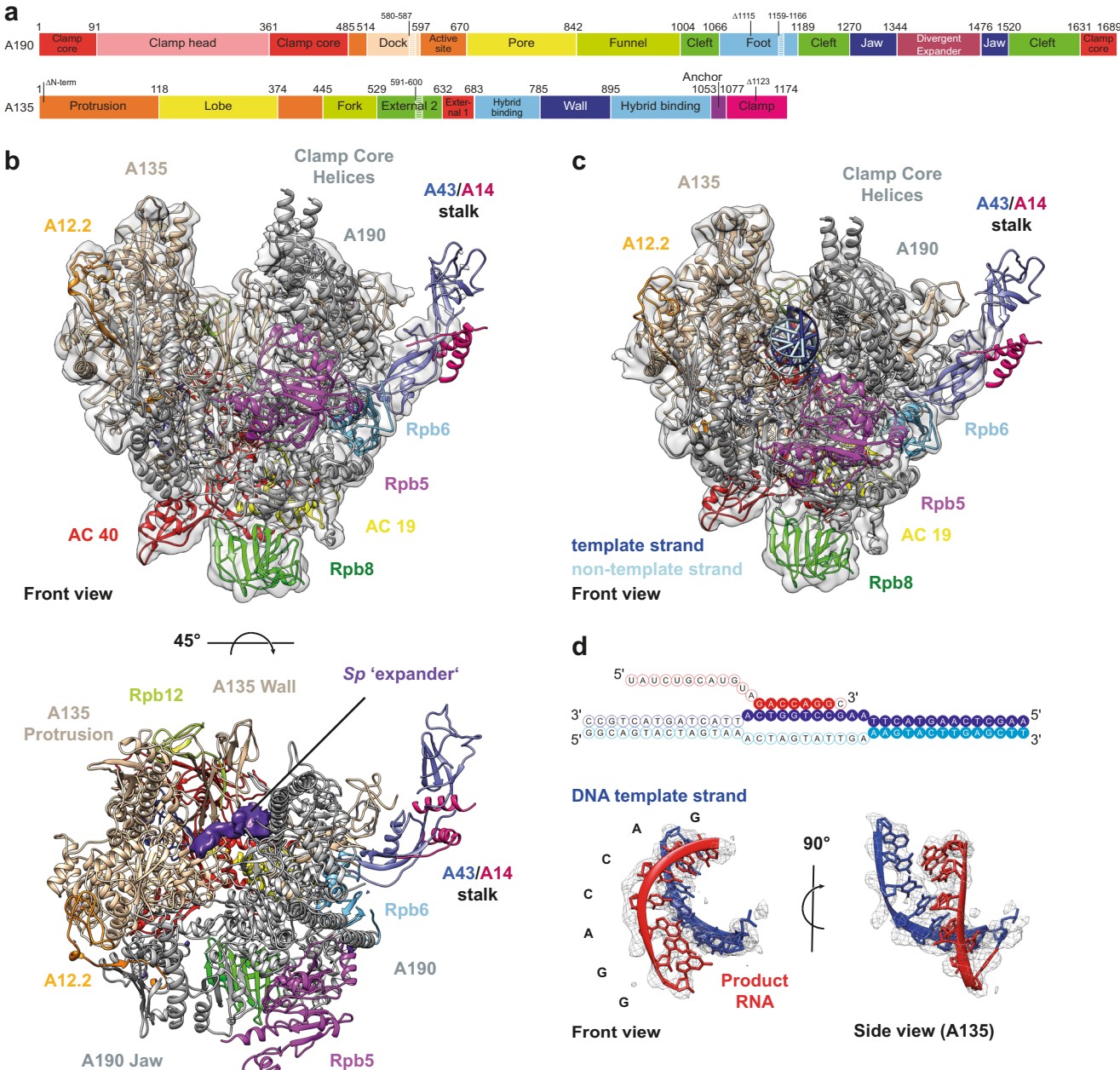

**Fig. 1 Structure of monomeric and elongating S. pombe Pol I. a** Sub-domain architecture of the Pol I subunits A190 and A135 in *Sp*. Insertion/Deletion of more than five residues compared to *Sc* are highlighted. **b** Cryo-EM reconstruction of monomeric *Sp* Pol I. Transparent, unsharpened cryo-EM density overlaid with ribbon model shows 12 subunits but lacks the A49/A34.5 subcomplex. The cleft is expanded and density (purple, space-filling) indicates the location of the 'expander' element. **c** Cryo-EM reconstruction of an *Sp* Pol I Elongation complex. The cleft is contracted and the expander is displaced by the hybrid. **d** Schematic representation of the artificial bubble construct used to establish an EC. Nucleotide bases included in the EC-density are highlighted. A model of the template-DNA/product-RNA hybrid overlaid with sharpened EC density is shown, the 3' nucleotide of the RNA is present but shows poor density and was thus not modeled.

buffer (Supplementary Fig. 5). In a high-salt buffer, about 32% of identified particles (49% of Pol I molecules) appear to be in a dimeric state as indicated by unsupervised 2D-classification. In contrast, dimer-prevalence is below 2% of particles in low-salt conditions. In line with this, DLS using the Prometheus Panta technology (Nanotemper) indicates an increase of hydrodynamic particle radius with increasing *Sp* Pol I concentration (Fig. 2b). Taken together, we concluded that reversible, connector-independent dimerization of *Sp* Pol I is possible. To understand the underlying molecular principles, we reconstructed the dimer architecture from its cryo-EM density.

**Inactive *Sp* Pol I dimers arrange in a divergent architecture.** A three-dimensional reconstruction of *Sp* Pol I dimers at 4.5 Å resolution was obtained in C2 symmetry (Fig. 3; Supplementary Table 1). Unambiguous placement of two *Sp* Pol I monomers and rigid body fitting of subdomains yielded a model of the *Sp* Pol I dimer. Whereas establishment of Pol I dimers depends on stalk-insertion into the cleft of a second monomer in *Sc*, the *Sp* dimers form by interaction of the stalk with the protrusion domain of subunit A135 in the other monomer (Fig. 3). A second interface forms between the 'dock domain' in subunit A190 of one monomer (including the *Sp* specific insertion of residues 580–587

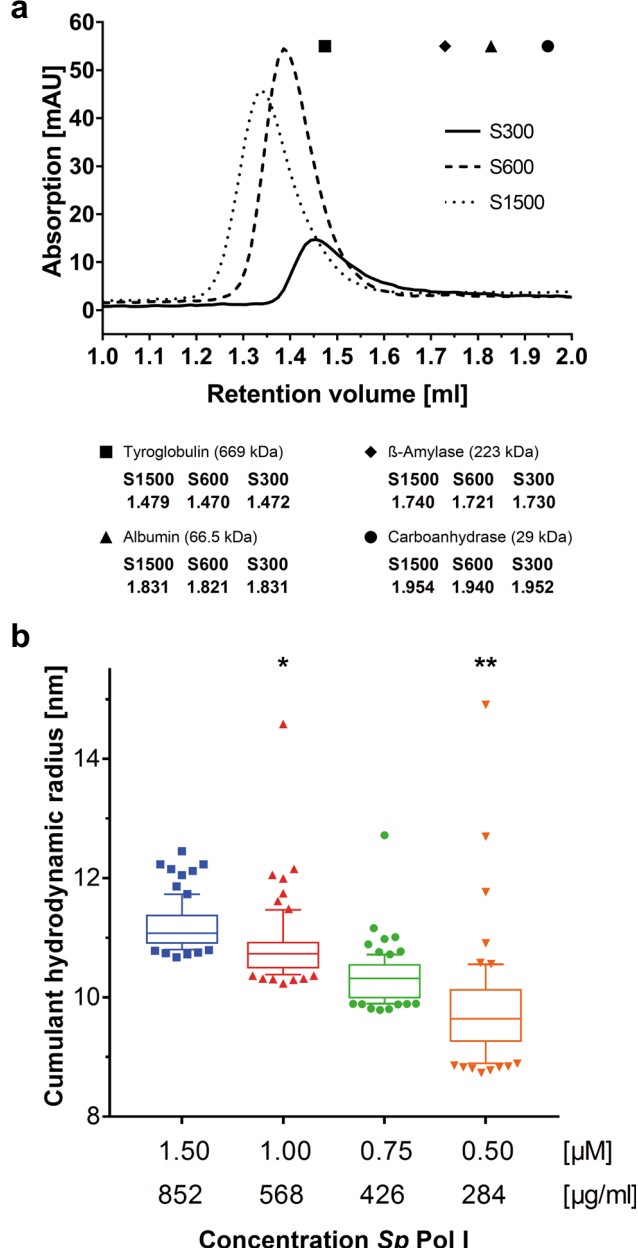

**Fig. 2 *Sp* Pol I can dimerize in vitro dependent on protein concentration and buffer condition. a** Analytical Size Exclusion Chromatography (SEC) of *Sp* Pol I in buffers containing 1500 mM (S1500; dotted), 600 mM (S600; dashed) or 300 mM (S300; continuous) Potassium Acetate. The main peak (A$_{280nm}$) elutes at higher apparent molecular weight in the high salt buffer. The average elution peaks of reference proteins are indicated: black square – Tyroglobulin; Black diamond – ß-Amylase; Black triangle – Albumin; Black dot – Carboanhydrase (exact retention volumes at the bottom). Compare Sup. Fig. 5 for negative-stain EM characterization of S300 and S1500 peak fractions and methods for details. **b** Dynamic Light Scattering (DLS) analysis of *Sp* Pol I particles (S600 buffer) measured using the Panta technology (Methods). Each mark represents an individual measurement ($n = 80$ for each Pol I concentration). At higher Pol I concentration, the hydrodynamic radius of particles in solution increases, indicating a shift of equilibrium towards dimerization. The boxes show the median (horizontal line inside box) and span from the 25$^{th}$ to the 75$^{th}$ percentiles, the whiskers reach up to the 90$^{th}$ and down to the 10$^{th}$ percentile, measurements beyond these are shown as individual points (* one outlier at 32.72 nm is not displayed; ** two outliers at 19.98 nm and 21.04 nm are not displayed).

and the Pol I specific helix α12a) with the wall subdomain of subunit A135, subunit AC40 and the common subunit Rpb12 of the second monomer. These two interfaces allow a tight association via the establishment of contacts between both Pol I upstream-faces independent of the connector element but utilizing organism-specific regions. Low-resolution density on the lobe of subunit A135 in both molecules in the dimers may indicate the presence of A49/A34.5 (Supplementary Fig. 6). This is in line with the suggestion that dissociation of this subcomplex is independent of Pol I inactivation in *Sp*[22].

**Cleft contraction is common to active Pol I.** Comparative structural modeling reveals that, similar to *Sc* Pol I reconstructions, the cleft is extended in monomers and contracted in ECs (Fig. 4; Supplementary Movie 1). Consequently, the 'bridge' helix of subunit A190 is mostly disordered in monomers (poor density despite higher overall resolution) and forms a well-structured helix in ECs (Supplementary Fig. 3e). The 'hinges' identified by comparing Pol II structures with Pol I dimers in crystals[5,6,17] are similar for *Sp* Pol I cleft modulation. Thus, polymerase activation by contraction is conserved between *Sc* and *Sp*. Further cleft expansion in *Sc* Pol I dimers likely results from mutual stalk-cleft insertion (Fig. 4b) and is not observed in *Sp* Pol I dimers (Fig. 4c; Supplementary Movie 2). Despite this divergence, functional importance is apparently conserved: Both molecules in a dimer are likely unable to initiate transcription since binding of the initiation factor Rrn3 to the stalk-dock region is blocked by the neighboring monomer. This Rrn3-binding is necessary for promoter recruitment in *Sc*[10,16,23–25], and likely conserved in *Sp*[22,26,27].

## Discussion

In this work, we described three single particle cryo-EM reconstructions of *Sp* Pol I, representing the only structures of this enzyme from an organism other than *Sc* to date. Availability of a second in vitro transcription system allows cross-validation of structural and functional investigations in future studies. Whereas we find that the general architecture of Pol I is conserved, structural details vary among organisms.

In contrast to *Sc*, density for the expander element was observed in *Sp* Pol I monomers, hinting at an additional possibility to prevent unspecific DNA-binding to Pol I monomers. Absence of cryo-EM density for the A49/A34.5 heterodimer may result from a loss of the subcomplex due to stress on the air-buffer-interface during freezing[28], a flexible association in *Sp*, or may indicate functional relevance. The A49/A34.5 subunit-complex plays an important role in Pol I initiation by stabilizing the non-template strand during promoterDNA melting and may support promoter escape in *Sc*[12,29]. The subcomplex shows similarities to initiation factors TFIIF and TFIIE in the Pol II transcription system, as suggested from homology to crystal structures and native mass spectrometry analysis[30]. However, the A49/A34.5 subcomplex has also been described as important for Pol I elongation[20,31,32] and is constantly attached to the Pol I core throughout elongation in vivo[33]. Dissociation of the subcomplex from *Sc* Pol I was observed under specific experimental conditions in vitro[20] or in ECs established using a nucleotide analogon[21]. Notably, the EC scaffold used in this study contains modified RNA which may lead to the loss of A49/A34.5[21], even though this was not the case in an *Sc* Pol I EC structure determined with the identical scaffold[15]. Taken together, it can be speculated that the A49/A34.5 subcomplex not only carries out multiple functions in Pol I initiation and elongation (as recently reviewed[34]), but also that these roles may vary in importance

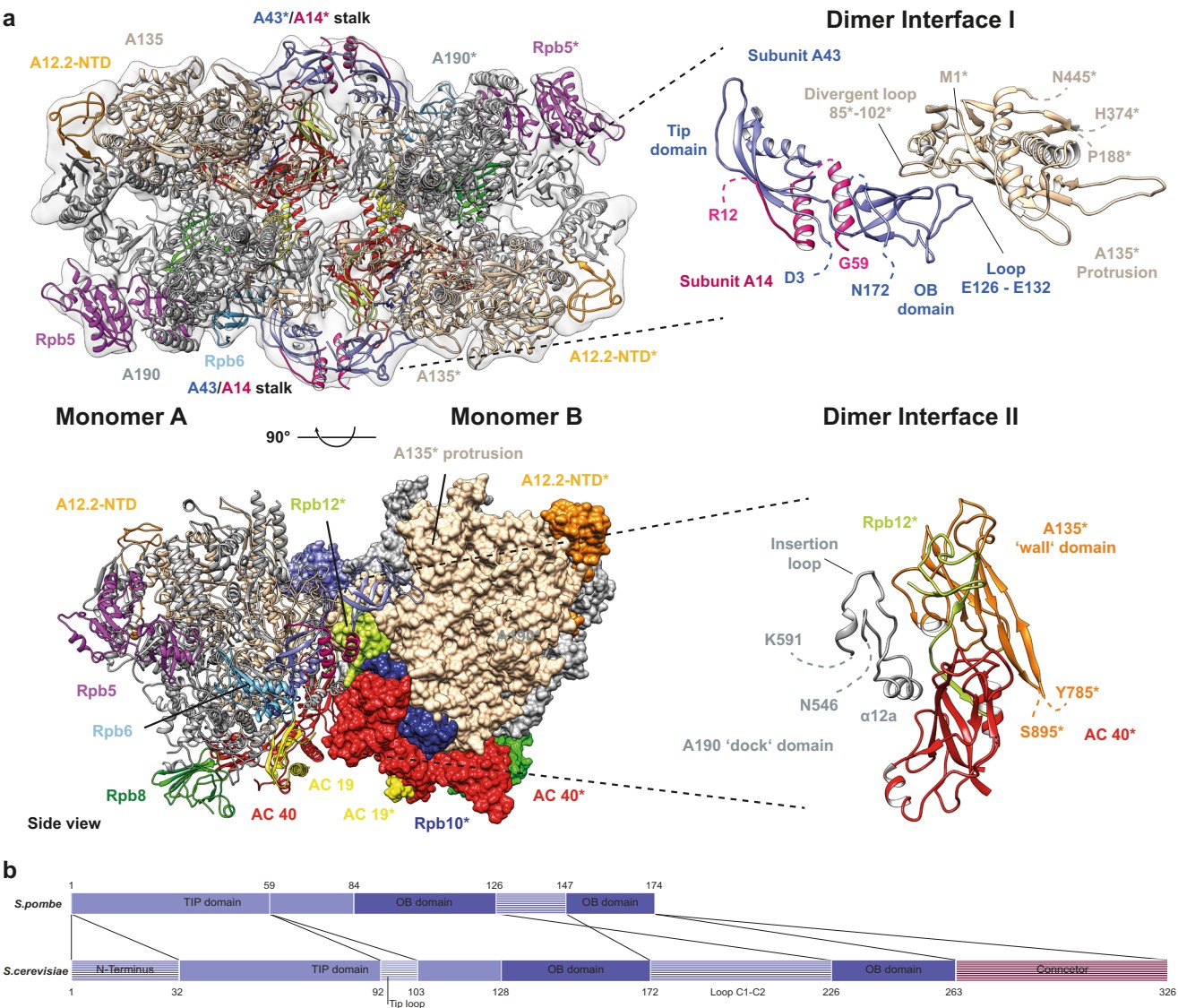

**Fig. 3 The architecture of *Sp* Pol I dimers diverges from its *Sc* counterpart. a** Cryo-EM reconstruction of dimeric *Sp* Pol I (transparent, unsharpened density). Dimers form by interaction of two monomers via two interfaces (right) on their upstream faces. Subunits of monomer B (ribbon in top panel, space-filling at the bottom) marked by asterisks. Interfaces are highlighted on the right and are largely composed of *Sp*-specific segments such as divergent loops in the protrusion of subunit A135, the OB-fold of subunit A43 and the dock domain of subunit A190. Some additional density in the back-region of both monomer clefts likely relates to the expander element. **b** Sub-domain architecture comparison of Pol I subunit A43 between *Sp* and *Sc* reveals the lack of a connector and a divergent C1-C2 loop. This loop is located at the distal end of the stalk sub-complex and involved in dimer-formation (compare panel a) by contacting the protrusion of the neighboring polymerase.

among organisms, even though functional complementation of *Sc* and *Sp* A49/A34.5 was previously shown[22,35].

Comparison of Pol I cleft expansion states showed that transcription activation by active center cleft contraction is conserved in *Sc* and *Sp*. This contraction distinguishes the Pol I transcription system from its Pol II/III counterparts[2,36]. Strikingly, well-defined dimers are present in *Sp* Pol I preparations that form independent of the A43 connector domain, which is required for dimerization in *Sc*[16]. Dimer prevalence in vitro is influenced by buffer conditions and Pol I concentration in a dynamic equilibrium in both organisms and not an artefact of chemical crosslinking or crystallization. We find that different organism-specific regions are involved in formation of inactive dimers in both, *Sp* and *Sc* Pol I, occluding the binding site for Rrn3 in both organisms. Hence, conserved Rrn3-association[23,37,38] can likely stabilize monomers as required for activation[39]. Regulated binding of Rrn3[40–43] then allows promoter recruitment and DNA-melting to take place[29,44].

Apparently, dimerization of Pol I molecules is possible but involves divergent primary and secondary structure elements. Dimers adopt a variable architecture, while their formation still results in a hibernating state. In vivo relevance of these findings and the reason for the dynamic association of the A49/A34.5-related subunits are still under debate awaiting further investigation. Nevertheless, we conclude that the principles underlying Pol I regulation 'activation by contraction' and 'hibernation by dimerization' are conserved among organisms.

## Methods

**Construction of AC40-tagged *S. pombe* strain.** A construct for genomic insertion of a 10xHis/Flag tag was ordered as plasmid (Gene Art). The construct was amplified and genomically inserted into the haploid *S. pombe* strain 972h-: A 100 ml YPD culture was started at optical density (OD_600) of 0.25 from an over night culture at 30 °C. After 5–6 h, OD_600 was at 1.0 and cells were harvested in 250 ml conical tubes (1361 *g*, 5 min). Cells were resuspended in 25 ml sterile water by vortexing and again centrifuged. Cells were resuspended in 1 ml of sterile 100 mM

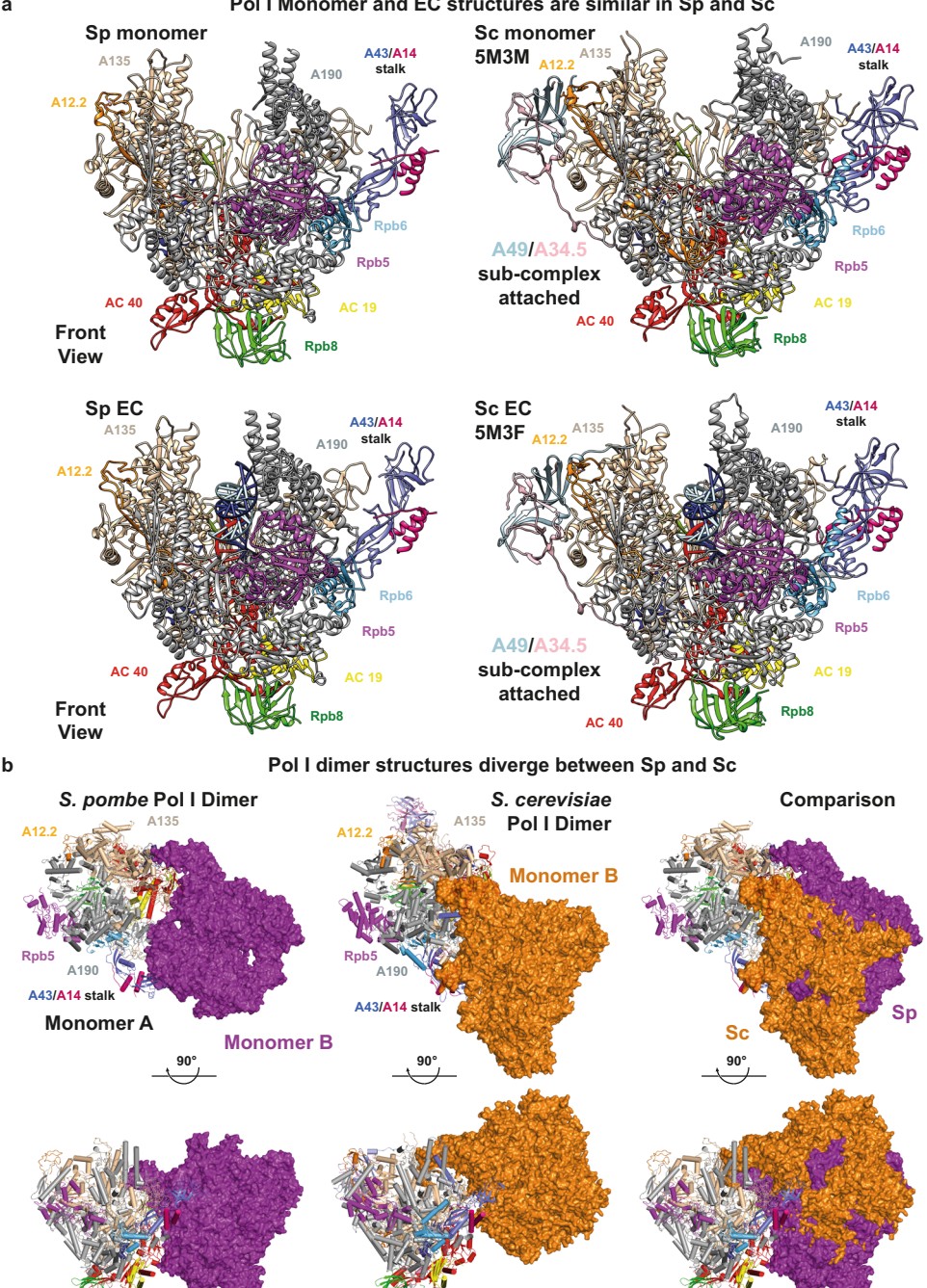

**Fig. 4 Comparison of *Sc* and *Sp* Pol I structures. a** Structural comparison of Pol I monomers (top) and ECs (bottom) in *Sp* (left) and *Sc* (right) displays the similar architecture of both enzymes. Upon EC formation, both Pol I versions contract their active sites (Supplementary Movie 1). **b** Comparison of *Sp* and *Sc* Pol I dimers. First monomer (cartoon tubes) overlaid via subunit A135. Second monomer (space-filling; *Sp* purple, *Sc* orange) is attached via upstream face but globally shifted in location. Increased cleft expansion compared to monomers is observed in *Sc* but not in *Sp*. This is likely a consequence of stalk insertion into the active center cleft and formation of the connector helix at the clamp core domain of the second monomer in *Sc*. Compare Supplemental Movie 2. **c** Quantification of cleft widths in *Sc* and *Sp* Pol I structures. In *Sc* structures, upstream width measured between residues Arg 434 in subunit A135 and Val 418 in subunit A190, downstream between Gly 231 and Lys 1331 in subunit A190. In *Sp* structures, upstream width measured between the corresponding residues Arg 409 in subunit A135 and Arg 425 in subunit A190, downstream between Lys 226 and Ser 1338 in subunit A190.

Li$_2$Ac solution. The suspension was then transferred into a 1.5 ml reaction tube and centrifuged for 15 s (tabletop centrifuge, full speed). Supernatant was removed and the pellet resuspended in 400 μl of fresh 100 mM Li$_2$Ac. In parallel, 500 μl of salmon sperm DNA (2 mg/ml) were boiled at 95 °C for 5 min and quickly chilled on ice. The cells were then split into 100 μl aliquots, pelleted and the supernatant removed. To a pellet, the following transformation mix was added in the following order: (1) 240 μl sterile PEG3350 (50% w/v), (2) 36 μl 1 M Li$_2$Ac, (3) 50 μl salmon sperm DNA (2 mg/ml), and (4) 34 μl PCR product of the insertion construct. Tubes were vigorously vortexed for more than 1 min and incubated at 30 °C for 30 min under shaking. Subsequently, reactions were transferred to 42 °C and incubated under shaking for 25 min. Cells were then pelleted (tabletop centrifuge at 6000 g for 15 s), the supernatant removed and cells were resuspended in 1 ml YPD medium. Cells were transferred into 15 ml conical tubes and shaken at 30 °C for 3 h. After centrifugation at 1361 g for 5 min, the pellet was resuspended in 500 μl sterile water and plated on YPD plates with Kanamycin/G418. The plates were incubated at 30 °C for 3-4 days, single colonies picked and re-plated on fresh plates. For verification of correct genomic insertion, the respective regions were amplified by PCR and sequenced.

**Fermentation of S. pombe**. S. pombe cells were plated on YPD plates and grown at 30 °C for 48–72 h. A preculture of 500 ml was started and grown over night in YPD at 30 °C under shaking. Cells were inspected for contaminations via light microscopy and secondary cultures of 2 l each were inoculated at a starting OD$_{600}$ of 0.3–0.5. After 10–12 h, cells were inspected visually and transferred into the 200 l fermenter at a starting OD$_{600}$ of 0.30–0.35. YPD medium was prepared in the fermenter, but pH was not adjusted and was therefore at ~6.0 initially. The medium was autoclaved and Ampicillin and Tetracycline were added to final concentrations of 100 μg/μl and 12.5 μg/μl, respectively. Antifoam reagent was added to reduce foaming during the fermentation. The fermenter was operated at 22 Nl/min (normal litres per minute) air influx and with 250 rpm stirring at 30 °C. After 11–13 h, an OD$_{600}$ of 6.0 to 7.5 was reached and cells were harvested with a continuous-flow centrifuge, resuspended in freezing buffer (150 mM HEPES pH 7.8, 60 mM MgCl$_2$, 20% v/v glycerol, 5 mM DTT, 1 mM PMSF, 1 mM Benzamidine, 60 μM Leupeptin, 200 μM Pepstatin; 0.5 ml buffer for each g of cells) and flash-frozen in liquid nitrogen for storage at −80 °C.

**Pol I purification**. The protocol for the purification of Sc Pol I[5,45] was slightly modified to be applicable for 10x His tagged S. pombe Pol I:

Frozen fermenter pellets (=150 g cells in a total volume of 225 ml) were thawed and ammonium sulfate concentration adjusted to 400 mM. Cells were lysed after adding 3 ml PI (100x) and 200 ml glass beads (diameter 0.5 mm) by bead beating for 90 min (30 s mixing, 60 s break) under constant cooling. After cell lysis glass beads were removed by filtering and washed with dilution buffer (100 mM HEPES pH 7.8, 20 mM MgCl$_2$, 400 mM (NH$_4$)$_2$SO$_4$). The crude cell extract was then centrifuged (4 °C; 8,600 g; JLA 16.250) for 60 min to remove the cell debris. The supernatant was afterwards ultracentrifuged (4 °C, 167,424 g; 45Ti rotor) for 90 min. The top fat layer was carefully removed using a 25-ml pipette, the mid-layer was subsequently collected without disturbing the viscous bottom DNA-pellet. The aspired mid-layer was dialysed overnight (16 h +) at 4 °C against dialysis buffer (50 mM KAc, 20 mM HEPES pH 7.8, 1 mM MCl$_2$, 10 % v/v glycerol, 10 mM ß-Mercaptoethanol, 1x PI (Benzamidine & PMSF)). The dialysed extract was ultracentrifuged for 2 h (4 °C; 41,856 g; 45Ti rotor). The Pol I containing pellet was resuspended and pellets pooled in Res/W1 buffer (1.5 M KAc, 20 mM HEPES pH 7.8, 1 mM MgCl$_2$, 10 mM Imidazole, 10 % v/v glycerol, 10 mM ß-Mercaptoethanol, 0.5 PI). After 2 h incubation on a rotating wheel (4 °C; 10 rpm) 4 ml equilibrated Ni-NTA beads were added to the suspension and further incubated for 4 h (4 °C, 7 rpm). After incubation the suspension was decanted into gravity columns, the Pol I binding Ni-NTA beads were subsequently washed with Res/W1 buffer (5 CV) and W2 buffer (300 mM KAc, 20 mM HEPES pH 7.8, 1 mM MgCl$_2$, 25 mM Imidazole, 10 % v/v glycerol, 10 mM ß-Mercaptoethanol) (5 CV). Pol I was then eluted with 20 ml total volume of E200 buffer (300 mM KAc, 20 mM HEPES pH 7.8, 1 mM MgCl$_2$, 200 mM Imidazole, 10 % v/v glycerol, 10 mM ß-Mercaptoethanol).

The eluate was therefore ultracentrifuged (4 °C; 46,378 g; 45Ti rotor) for 20 min and loaded on a MonoQ 10/100 column (GE Healthcare) equilibrated with 15% B (Mono-Buffer A: 20 mM HEPES pH 7.8, 1 mM MgCl$_2$, 10% v/v glycerol, 5 mM DTT; Mono-Buffer B: 2 M KAc, 20 mM HEPES pH 7.8, 1 mM MgCl$_2$, 10% v/v glycerol, 5 mM DTT). Pol I was eluted with a linear gradient of 13 CVs from 0.3 M to 1.4 M KAc (elution at around 0.9 M KAc). Pol I containing fractions were pooled and diluted 200 mM KAc with Buffer A and again centrifuged (4 °C; 16,696 g; 45Ti rotor). Next, the sample was loaded on a MonoS 5/50 column (GE Healthcare) equilibrated with 200 mM KAc. Pol I was eluted with a linear gradient from 0.2 M to 0.7 M KAc with a plateau of 5 CV at 0.35 M (elution at around 0.5 M KAc). The peak fractions were analyzed on a gel, pooled, concentrated (Amicon; 100 kDa Molecular weight cut-off), flash-frozen in liquid nitrogen, and stored at −80 °C.

**RNA elongation and cleavage assays**. Purified Sc or Sp Pol I (1, 0.5 or 0.25 pmol) was pre-incubated with 0.25 pmol of pre-annealed minimal nucleic acid scaffold (template DNA: 5′-CGAGGTCGAGCGTGTCCTGGTCTAG-3′, non-template

DNA: 5′-CGCTCGACCTCG-3′; RNA: 5′-FAM-GACCAGGAC-3′) in transcription buffer (20 mM HEPES pH 7.8, 60 mM (NH$_4$)$_2$SO$_4$, 8 mM MgSO$_4$, 10 μM ZnCl$_2$, 10% (v/v) glycerol, 10 mM DTT) for 20 min at 20 °C. For RNA elongation, NTPs (1.4 mM end concentration each) were added and the reaction was incubated for 30 min at 28 °C. To examine cleavage activity, the pre-incubated reaction with a twofold molar excess of Pol I compared to scaffold was incubated for 30 min at 28 °C without the addition of NTPs. To stop the reaction an equal amount of 2x RNA loading dye (8 M Urea, 2× TBE, 0.02% bromophenol blue, 0.02% xylene cyanol) was added and the sample was heated to 95 °C for 5 min. As control 0.25 pmol of scaffold was treated identically, without the addition of polymerase and NTPs. 0.125 pmol of FAM-labeled RNA product (as well as a marker containing 9 nt, 15 nt and 21 nt long FAM-labeled RNAs: 5′-FAM-GACCAGGAC-3′, 5′-FAM-AACGGAGACCAGGAC-3′, 5′-FAM-UGUUCUUCUGGAAGUCCA GTT-3′) was separated by gel electrophoresis (20% polyacrylamide gel containing 7 M Urea) and visualized with a Typhoon FLA9500 (GE Healthcare).

**Preparation of Pol I elongation complex**. Synthetic DNA (IDT) and RNA (Qiagen) oligonucleotides were designed and assembled as described[15], with the scaffold sequence for the template DNA (5′-AAGCTCAAGTACTTAAGCCTGGT CATTACTAGTACTGCC-3′), non-template DNA (5′-GGCAGTACTAGTAAAC TAGTATTGAAAGTACTTGAGCTT-3′), and RNA (5′-UAUCUGCAUGUAGAC CAGGC-3′; for the underlined nucleotides a methylene bridge between the 2′-O and the 4′-C of the ribose ring has been formed, thus creating a locked nucleic acid, LNA). Annealing was achieved by equimolar mixing (40 μM), then heating to 95 °C, and gradually reducing the temperature to 20 °C over 90 min. Pol I (1 mg/ml) was incubated with a 1.35-fold molar excess of pre-annealed EC-scaffold for 30 min at room temperature.

**Crosslinking**. Purified Pol I (Mono S Eluate at concentration 1.0–1.3 mg/ml) was incubated with BS3 (1 mM final concentration) for 30 min (30 °C, 300 rpm), the reaction was stopped by adding Asp-Lys (9 mM final; 25 °C, 300 rpm) for 20 min followed by ammonium hydrogen carbonate (60 mM final; 25 °C; 300 rpm) for 20 min.

**Cryo-EM grid preparation**. The samples were centrifuged (4 °C; 21,130 g; Eppendorf tabletop centrifuge) for 5 min, to remove aggregates, and the supernatant carefully transferred into a fresh tube. The sample was then applied to a Superose 6 Increase 3.2/300 column in Solo4 buffer (5 mM HEPES pH 7.8, 1 mM MgCl$_2$, 10 μM ZnCl$_2$, 150 mM KCl, 5 mM DTT). The Pol I containing fraction was again centrifuged (4 °C; 21,130 g) for 5 min, and concentration was adjusted to approximately 100 μg/ml. Four μl of sample was applied to a glow discharged (2x; 0.4 mbar 15 mA; 100 s) R1.2/1.3 Cu #300 grid (Quantifoil) and plunge frozen in liquid ethane (Vitrobot Mark IV, Thermo Fisher Scientific; 100 % humidity; 4 °C; 5 s wait time; 5 s blotting time; blot force 12).

**Single-particle cryo-EM**. Images were collected on a Titan Krios Electron Microscope (Thermo Fisher Scientific) at 300 keV. Movies of 40 frames were acquired on a Falcon III direct electron detector at 75,000x magnification (pixel size 1.0635 Å). The movies were recorded in linear mode with a dose rate of ~19 e$^−$/px/s and a total dose of around 86 e$^−$/Å$^2$. The defocus span from −1.4 μm to −2.4 μm alternating in 0.2 μm intervals with a total of four exposures per hole.

**Data processing**. The EC dataset was processed using the RELION 3.0 suite[46] (Supplementary Fig. 2). Movie frames were aligned and dose weighted using Relion's own implementation of MotionCor and Contrast Transfer Function (CTF) parameters were estimated using GCTF. A total of 3,598 movies were chosen based on accumulated motion, visual inspection and CTF values, astigmatism, defocus and maximal resolution. A set of 100 randomly picked micrographs throughout the dataset was chosen for reference-free auto-picking using the Laplacian-of-Gaussian (LoG) routine and yielding 2,829 particles. Two-dimensional classification resulted in templates for reference-based auto-picking yielding 299,038 particles. Two-fold binned particles (128 pixel boxes) were subjected to reference-free 2D classification (250 Å mask). Following removal of contaminants, a total of 156,493 unbinned particles were selected and aligned in 3D using an initial model generated in RELION as reference. These particles then underwent CTF refinement, bayesian polishing, followed by another round of CTF refinement. Masked Auto-refinement resulted in a reconstruction at 3.89 Å overall resolution (0.143 FSC). Removal of particles showing increased flexibility of the Jaw and Clamp subdomains were removed by 3D Classification, resulting in 61,954 particles that allow reconstruction of an Sp Pol I EC at 4.00 Å resolution.

The 'monomer' dataset was processed using the RELION 3.0 suite[46] unless stated otherwise (Supplementary Fig. 1). After importing pre-averaged movie frames (sums) the Contrast Transfer Function (CTF) parameters were estimated with the embedded Gctf program. Pre-processing was performed as described for the EC dataset. LoG picking resulted in a total of 11,594 particles from which 1,142 were chosen for template-based auto-picking following 2D classification. Removal of ~90% of initially picked particles can be attributed to contamination and damage resulting from stress on the air-buffer interface. Initial auto-picking identified 874,753 particles from 4,333 micrographs, of which ~50% were discarded as

contamination based on 2D-Classification. A total of 477,791 particles were subjected to 3D classification using PDB 5M3M as reference. This allowed the removal of damaged particles and Pol I particles with highly flexible subdomains. The remaining 79,313 particles were subjected to CTF-refinement and bayesian polishing as for EC particles. The final 3D-reconstruction of monomeric *Sp* Pol I at a nominal resolution of 3.84 Å shows an even orientational distribution of particles and some flexibility in the peripheral jaw, clamp and stalk regions.

In initial 2D classifications, minor dimer-classes were noticed. Thus, auto-picked particles were re-extracted in larger boxes of 360 pixels and analyzed in a second, independent processing tree. From 510,315 Pol - like particles selected by 2D classification, a class of 17,552 particles could be attributed to well-defined dimers. Particles were centered on the interface between both Pols, re-extracted and another 450 poor particles removed based on 2D classification without sampling. Final 3D auto refinement imposing C2 symmetry yielded a reconstruction of *Sp* Pol I dimers at an overall resolution of 4.5 Å.

**Model building**. At nominal resolutions of 3.8–4.0 Å, we derive near-atomic models for most regions of *Sp* Pol I monomers and the EC. To commence model interpretation, we constructed homology models of subunits A190, A135, AC40, AC19, A43, A14 (ker1 in *Sp*) and A12.2 based on sequence comparison with their *Sc* homologs, alignment of actual and predicted secondary structures, and domain searches using HHPRED[47]. To construct homology models, the MODELLER[48] implementation of UCSF Chimera was used[49]. Structures of the general subunits Rpb5, Rpb6, Rpb8, Rpb10 and Rpb12 were imported from the crystal structure of *Sp* Pol II[50]. Subdomain boundaries were defined based on *Sc* homology (Fig. 1 and Supplementary Fig. 4). Subdomains were then rigid body fitted into EC densities (which were first obtained) using COOT[51]. While many regions allowed accurate fitting of sidechain orientations in the sharpened cryo-EM map, others suffered from poor main-chain tracing. Hence, density-guided modeling was performed in the clamp head, dock-insertion, foot-insertion and part of the jaw regions in subunit A190, as well as the subunit A12.2 and the toe domain of subunit AC40. Modeling of stalk-subunits A43 and A14 was limited to rigid body fitting of trimmed homology model domains in the unsharpened cryo-EM density. As a final step, real-space refinement was carried out using phenix.refine[52]. The *Sp* Pol I monomer was built by placement of the EC model and adjustment of subdomains in COOT, followed by manual inspection and real-space refinement using phenix.refine. The dimer model was constructed by placement of two monomers, rigid body fitting of subdomains and refinement in phenix-refine using NCS restraints.

**Concentration-dependent dimerization using dynamic light scattering**. Frozen *Sp* Pol I was thawed and diluted (1.5 μM, 1.0 μM, 0.75 μM, 0.5 μM) in S600 buffer (10 mM HEPES pH 7.8, 1 mM $MgCl_2$, 0.01 mM $ZnCl_2$, 5 mM DTT, 1.5 % (v/v) glycerol, and 0.6 M KAc) to a final volume of 20 μl. Technical duplicates of 10 μl of each Pol I concentration were loaded into glass capillaries (Prometheus NT.48 Series nanoDSF Grade Standard Capillaries) and each capillary mounted into a Prometheus Panta (NanoTemper Technologies GmbH). Fourty consecutive DLS measurements per capillary were taken at 75 % LED power, 100% Laser power and 15 °C (total measurements per condition; $n = 80$). Calculation and visualization were carried out using GraphPad Prism version 8.0.1 for Windows, GraphPad Software, La Jolla California USA, www.graphpad.com. The boxes extend from the 25th to 75th percentiles[53]. The whiskers in Fig. 2b are drawn down to the 10th percentile and up to the 90th. Points below and above the whiskers are drawn as individual points.

**Analytical size exclusion chromatography**. A total of 50 μg of frozen *Sp* Pol I was thawed and diluted to 2.93 μM with SEC buffer (5 mM HEPES pH 7.8, 1 mM $MgCl_2$, 10 μM $ZnCl_2$, 5 mM DTT, and 1.5 M KAc (S1500), 600 mM (S600), or 300 mM (S300)) to a total volume of 30 μl. The sample was centrifuged (4 °C; 21,130 g; Eppendorf tabletop centrifuge) for 5 min, to remove aggregates, and the supernatant carefully transferred into a fresh tube. The sample was then applied to a Superose 6 Increase 3.2/300 column (GE Healthcare; flow 0.035 ml/min; 50 μl fractions) in the respective buffer (S1500, S600, or S300). Peak fractions (Fig. 2) were diluted and negatively stained. After each run, a 30 μl mixture of marker proteins (Thyroglobulin (669 kDa), ß-Amylase (223 kDa), Albumin (66.5 kDa), Calmodulin (29 kDa) was applied to the column for calibration in each buffer.

**Negative staining, EM data collection and image processing**. Analytical SEC peak fractions were diluted to 20% (v/v) and 10% (v/v) in their respective buffers and were centrifuged (4 °C; 21,130 g; Eppendorf tabletop centrifuge) for 5 min. Five μl of the samples were then applied to 400-mesh copper grids (G2400C; Plano) with a self-made carbon film of ~7 nm thickness (self-made). After 30 s, grids were washed in 200 μl ddH₂O for 30 s, and stained three times in 20 μl saturated uranyl formate solution (30 s). After each step, excess liquid was removed with a filter paper. Images were collected on a JEOL 2100-F Transmission Electron Microscope operated at 200 keV and equipped with TVIPS-F416 (4kx4k) CMOS-detector at 40,000x magnification (pixel size 2.7 Å) with alternating defocus (−2.5 to −4.5 μm).

The images were processed using RELION 3.1 (see above). For the high-salt SEC peak fraction (S1500), a total of 90 out of 98 collected micrographs were analyzed. A set of 10 randomly chosen images was used to train and optimize the

reference-free auto-picking using Laplacian-of-Gaussian (LoG) routine. These settings were then applied on all 90 images yielding 40,544 particles that were applied to reference-free 2D classification (380 Å mask). After removal of junk, a total of 20,054 particles were classified into 16 classes.

For the shown low-salt SEC peak fraction (S300), 129 micrographs were analyzed. A set of 10 randomly chosen images was used to train and optimize the reference-free auto-picking using Laplacian-of-Gaussian (LoG) routine. These particles underwent selection based on 2D classification and 3D centering yielding 3,532 particles that were subsequently used as a template for a reference-based auto-picking from all 129 images resulting in 36,733 particles. After 3D centering using the filtered density of PDB 5M3M as reference and removal of junk particles by 2D classification (380 Å mask), 24,462 particles remained. The outcome of a 2D classification into 16 classes was then compared to the high-salt particles (Supplementary Fig. 5).

**Reporting summary**. Further information on research design is available in the Nature Research Reporting Summary linked to this article.

# Data availability

The cryo-EM density of *Sp* Pol I monomer, dimer and EC have been deposited in the Electron Microscopy Data Bank under accession codes EMD-11840, EMD-11841 and EMD-11842, respectively. Coordinates of the *Sp* Pol I monomer, dimer and EC were deposited with the Protein Data Bank under accession codes 7AOC, 7AOD and 7AOE, respectively. The data underlying Fig. 2 and unprocessed gel scans are provided in a separate Source Data File. Further material can be obtained from the corresponding author upon reasonable request. Source data are provided with this paper.

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

## Acknowledgements

We thank Michael Pilsl, Herbert Tschochner, Achim Griesenbeck and Philipp Milkereit for help and discussion, and Mona Höcherl for technical assistance. We thank Astrid Bruckmann for mass spectrometry, Gerhard Lehmann and Nobert Eichner for IT support, Gernot Längst for assistance with DLS measurements, Ralph Witzgall for JEM2100F access, and Achilleas Frangakis and Utz Ermel for initial cryo-EM screening. Cryo-EM data were collected at the cryo-EM facility of the University of Würzburg with support from Bettina Böttcher and Christian Kraft. This work was supported by Deutsche Forschungsgemeinschaft SFB 960 (TP-A8) and the Emmy-Noether Programm (DFG grant no. EN 1204/1-1 to CE).

## Author contributions

F.H. carried out *Sp* Pol I purification and characterization, prepared cryo-EM grids and carried out sequence analysis. F.H. and C.E. processed cryo-EM data. J.D. carried out functional elongation/cleavage assays. P.B. and C.E. built *Sp* Pol I models. C.E. designed and supervised research, established strains and purification protocols and prepared the manuscript with input from all authors.

## Funding

## Competing interests

The authors declare no competing interests.
