## [Peer Review File · Nature Communications]

REVIEWER COMMENTS

Reviewer #1 (Remarks to the Author):

This manuscript describes three structures of the *S. pombe* RNA Polymerase I in different states; two inhibited forms as monomer and dimer, and a monomeric elongation complex. These structures are contrasted with earlier work from *Sc* Pol I, revealing similar regulatory strategies, but significant differences in the molecular details. The resolution of a monomeric inhibited Pol I is especially novel, and indicates that dimerisation is not the sole reason for polymerase inhibition. Most notably, dimers of Pol I are resolved, but utilise entirely unique dimerisation interfaces compared with *cerevisiae* Pol I. Collectively, these structures are a significant advance in the Pol I field and are appropriate for publication in Nature Communications pending the resolution of a major issue described below.

Major Issue

It is possible that the dimerisation of *Sp* Pol I is induced by chemical crosslinking, as alluded to by the authors in the discussion. As always with crosslinking, artefacts are a risk so can the authors provide assurances or at least some reasoning as to why their cross-linked dimer structure is biologically relevant? Given the lack of conservation of the A43 Connector between *Sc* and *Sp*, this is important. In this regard, it is concerning/confusing that their sample wasn't subjected to size exclusion chromatography (preferably together with MALS) as a similar analysis was performed for the purification of the *Sc* Pol I crystal structure published by the same lead author in Nature (2013). I would urge the authors to include data that shows dimers can occur without crosslinking e.g. SEC-MALS analysis, and/or that the dimerisation interfaces in *Sp* Pol I are functionally important.

Minor Issues

The phrase on page 5 that dimerisation 'functionally disabling both molecules' needs clarification: how does dimerisation result in inactivation? By precisely the same mechanism that was described for *Sc* Pol I i.e. blocking the association with Rrn3 and Rrn7?

What is the basis for attributing the density within the cleft to the expander, given that it's (presumably) poorly connected to the rest of A190 and is not detailed enough to be built? Whilst I don't disagree that it's likely to be the expander, I think it could be better justified, perhaps by comparison to the *Sc* Pol I expander. Assuming they occupy the same relative position in both Pols I, and have conserved sequences (which seems apparent from Supplemental Figure 4, pt 1/4), this could be much more compelling.

Reviewer #2 (Remarks to the Author):

The manuscript by Heiss et al. reports cryo-EM structures of RNA polymerase I (Pol I) from *S. pombe* in three different functional states, i.e. elongation complex (EC), free monomeric and free dimeric, at resolutions ranging from 3.8 to 4.5 Å. Comparison of the free monomeric and EC structures shows that the DNA-binding cleft contracts upon interaction with the transcription bubble. As observed for *S. cerevisiae* Pol I, *S. pombe* Pol I dimerizes but the dimer interface significantly differs from that observed in the *S. cerevisiae* enzyme. From their results, the authors conclude that activation by cleft contraction and hibernation by dimerization are conserved mechanisms in spite of divergences.

Overall, the work is technically sound and interesting for the field, but a few issues should be addressed before it is ready for publication.

A significant finding is the fact that *S. pombe* Pol I dimerizes in solution while lacking the A43-Ct

(connector), necessary for *S. cerevisiae* Pol I dimerization. However, the authors' statement that hibernation by dimerization represents a 'conserved regulatory mechanism' (end of Abstract) would require further evidence. One way to support their claim would be to use diploids harboring different tags on each allele of a specific subunit, then crosslink under various *in vivo* conditions and co-IP with one of the tags to show presence of the other tag in conditions such as starvation (where *S. cerevisiae* Pol I was shown to dimerize *in vivo*). An alternative would be to produce deletion mutants at the dimerization interface (for example, by deleting the very tip of A43, as judged from Figure S5) and show they do not dimerize in biochemical conditions where the wild-type enzyme does. This mutant should also show higher Pol I promoter levels than the wild-type enzyme upon starvation.

The complete lack of density for the A49/A34.5 subcomplex is of potential functional interest. This is most surprising in the DNA-free Pol I structure, as the sample contains A49 and A34.5 according to SDS-PAGE and MS (page 3) and it is crosslinked before grid preparation (page 3). The authors argue that the subunits may be lost due to stress at the air-buffer interface (page 5) but, are the subunits still present in the complex after crosslinking as determined by MS? Understanding the origin of their result is important for the discussion. In case the subcomplex is indeed very labile, what is the relevance of this finding *in vivo*? The authors should perform ChIP on one or both subunits in the subcomplex to assess its presence along the transcribed region.

In addition, the EC structure shows weak density that may be attributed to A12-Ct. A similar position for this domain was observed for *S. cerevisiae* Pol I EC in the presence of GMPCPP (Tafur et al., eLife 2019;8:e43204). A specific comparison between these structures, including additional figure/panels, should be made. Is the position of the A12-Ct in the Tafur paper coherent with the weak density in the EC structure? The authors should also discuss on the functional relevance of this observation and the possible connection with their nucleic acid scaffold, which includes three locked nucleic acid bases at the 3' end of the RNA primer.

In general, the manuscript would greatly benefit from a more detailed presentation of results and its discussion, at least for the following sections:

- The conservation of cleft contraction upon nucleic acid binding in the cleft is one of the main observations, but its description and corresponding discussion in the manuscript is succinct. What is the extent of the cleft movement? How does it compare to *S. cerevisiae* Pol I? Are the hinge regions conserved between the enzymes? Figure S3e should be improved by showing cleft width distances, hinge points, superposition of the pombe and cerevisiae enzymes, etc.
- Related to this, helix H0 in *S. cerevisiae* Rpb6 is ordered only in Pol I and not in Pol II or Pol III. This has been connected to increased stalk attachment to the Pol I core and to rigidity of the clamp-shelf module. The authors should describe and discuss whether this is the same in *S. pombe* Pol I. Moreover, the authors should include an alignment of Rpb6 (and also the other 4 subunits shared by the three nuclear RNA polymerases) in Figure S4. Figure S4 should also indicate relevant structural domains discussed in the main text.
- Conformational changes between the free monomeric and dimeric states of the enzyme are also poorly described. In *S. cerevisiae*, further cleft expansion is observed upon dimerization, which associates with binding of the expander loop inside the cleft. Is further cleft expansion and expander loop location conserved in *S. pombe* Pol I? A figure/panel on such conformational changes in *S. pombe* should be added.
- Finally, the main text contains only 2 figures. Figures S3 and S5 are very relevant and should be moved into the main text.

Reviewer #1 (Remarks to the Author):

This manuscript describes three structures of the *S. pombe* RNA Polymerase I in different states; two inhibited forms as monomer and dimer, and a monomeric elongation complex. These structures are contrasted with earlier work from *Sc* Pol I, revealing similar regulatory strategies, but significant differences in the molecular details. The resolution of a monomeric inhibited Pol I is especially novel, and indicates that dimerisation is not the sole reason for polymerase inhibition. Most notably, dimers of Pol I are resolved, but utilise entirely unique dimerisation interfaces compared with *cerevisiae* Pol I. Collectively, these structures are a significant advance in the Pol I field and are appropriate for publication in *Nature Communications* pending the resolution of a major issue described below.

Major Issue

It is possible that the dimerisation of *Sp* Pol I is induced by chemical crosslinking, as alluded to by the authors in the discussion. As always with crosslinking, artefacts are a risk so can the authors provide assurances or at least some reasoning as to why their cross-linked dimer structure is biologically relevant? Given the lack of conservation of the A43 Connector between *Sc* and *Sp*, this is important. In this regard, it is concerning/confusing that their sample wasn't subjected to size exclusion chromatography (preferably together with MALS) as a similar analysis was performed for the purification of the *Sc* Pol I crystal structure published by the same lead author in *Nature* (2013). I would urge the authors to include data that shows dimers can occur without crosslinking e.g. SEC-MALS analysis, and/or that the dimerisation interfaces in *Sp* Pol I are functionally important.

We appreciate the feedback and have addressed the question using two experimental strategies: We carried out calibrated analytical SEC in different buffers conditions showing that *Sp* Pol I elution strongly depends on salt concentration. High salt concentration favours an early elution which we associate with increased dimer formation by negative stain EM analysis. This is independent of crosslinking. Furthermore, using the Prometheus Panta dynamic light scattering technology (Nanotemper), we show that the hydrodynamic radius of particles increases with Pol I concentration, indicating a shift of equilibrium towards dimer prevalence similar to *Sc* Pol I as mentioned by reviewer #1 (Engel et al. *Nature* 2013). Results are now included in the main manuscript with a new figure.

Minor Issues

The phrase on page 5 that dimerisation 'functionally disabling both molecules' needs clarification: how does dimerisation result in inactivation? By precisely the same mechanism that was described for *Sc* Pol I i.e. blocking the association with Rrn3 and Rrn7?

We added a section to the discussion that focuses on the issue. While both, Rrn3 and Rrn7 are probably unable to bind, the conserved factor Rrn3 is most prominently occluded based on similarities between *Sp* and *Sc* described in the literature.

What is the basis for attributing the density within the cleft to the expander, given that it's (presumably) poorly connected to the rest of A190 and is not detailed enough to be built? Whilst I don't disagree that it's likely to be the expander, I think it could be better justified, perhaps by comparison to the *Sc* Pol I expander. Assuming they occupy the same relative position in both Pols I, and have conserved sequences (which seems apparent from Supplemental Figure 4, pt 1/4), this

could be much more compelling.

We have added the requested figure and have further clarified the putative nature of the assignment in the main text. It is indeed rather interesting that expander-mediated monomer inactivation is observed in *Sp* but not *Sc*. We included a section in the discussion but cannot provide a clear reason for this observation.

Reviewer #2 (Remarks to the Author):

The manuscript by Heiss et al. reports cryo-EM structures of RNA polymerase I (Pol I) from *S. pombe* in three different functional states, i.e. elongation complex (EC), free monomeric and free dimeric, at resolutions ranging from 3.8 to 4.5 Å. Comparison of the free monomeric and EC structures shows that the DNA-binding cleft contracts upon interaction with the transcription bubble. As observed for *S. cerevisiae* Pol I, *S. pombe* Pol I dimerizes but the dimer interface significantly differs from that observed in the *S. cerevisiae* enzyme. From their results, the authors conclude that activation by cleft contraction and hibernation by dimerization are conserved mechanisms in spite of divergences.

Overall, the work is technically sound and interesting for the field, but a few issues should be addressed before it is ready for publication.

A significant finding is the fact that *S. pombe* Pol I dimerizes in solution while lacking the A43-Ct (connector), necessary for *S. cerevisiae* Pol I dimerization. However, the authors' statement that hibernation by dimerization represents a 'conserved regulatory mechanism' (end of Abstract) would require further evidence. One way to support their claim would be to use diploids harboring different tags on each allele of a specific subunit, then crosslink under various *in vivo* conditions and co-IP with one of the tags to show presence of the other tag in conditions such as starvation (where *S. cerevisiae* Pol I was shown to dimerize *in vivo*). An alternative would be to produce deletion mutants at the dimerization interface (for example, by deleting the very tip of A43, as judged from Figure S5) and show they do not dimerize in biochemical conditions where the wild-type enzyme does. This mutant should also show higher Pol I promoter levels than the wild-type enzyme upon starvation.

In order to further investigate *Sp* Pol I dimerization *in vitro*, we tested buffer and concentration dependence in solution as also requested by reviewer #1 in the absence of crosslinker and included the results in the main manuscript. We conclude that dimerization can take place *in vitro* and is stabilized by chemical crosslinking.

We agree that the presented results are not sufficient to demonstrate that hibernation by dimerization and activation by contraction of Pol I are highly conserved regulatory mechanisms. However, based on our results we would like to propose that these concepts may be used of relevance in different organisms. This is reflected by careful wording: '... suggest conservation of regulatory mechanisms among organisms...'

Modification of the dimer interface is complicated, since contacts are in part located in common subunits that are also incorporated in Pols II and III. On the other hand, mutation of elements in Pol-I-specific subunits may result in Pol-I-complex instability due to folding issues or other off-target effects. However, we aimed at analyzing the effect of the stalk as a whole on dimerization and succeeded in creating an A14-deletion strain carrying a tag on subunit AC40. The strain cannot

synthesize a normal Pol I stalk and shows a strong growth defect but also reduced Pol I purification yields. This indicates that dimer formation is not essential *in vivo* (similar to *S. cerevisiae* as shown by Torreira et al. 2017, eLife). Analysis of delta-stalk Sp Pol I is currently incomplete and thus not included in this revision. Follow-up experiments will help to elucidate the *in vivo* relevance of the findings presented here.

The complete lack of density for the A49/A34.5 subcomplex is of potential functional interest. This is most surprising in the DNA-free Pol I structure, as the sample contains A49 and A34.5 according to SDS-PAGE and MS (page 3) and it is crosslinked before grid preparation (page 3). The authors argue that the subunits may be lost due to stress at the air-buffer interface (page 5) but, are the subunits still present in the complex after crosslinking as determined by MS? Understanding the origin of their result is important for the discussion. In case the subcomplex is indeed very labile, what is the relevance of this finding *in vivo*? The authors should perform ChIP on one or both subunits in the subcomplex to assess its presence along the transcribed region.

We agree that the loss of A49/A34.5 is somewhat curious. In order to carry out ChIP on different regions of the rDNA gene, we aimed at tagging subunits A135, A49 and A34.5 in Sp. Unfortunately, we did not succeed in creating the required strains in the timeframe of the revision. Nevertheless, we will continue to pursue this line of investigation. It should be noted that a comparable loss of the A49/A34.5 subcomplex is in some cases observed in Sc Pol I *in vitro* (Tefur et al, Elife 2019) even though ChIP shows a rather consistent association over the gene *in vivo* (Beckouet et al., MCB 2008). Correlating our observed loss (or flexibility) of the subcomplex with results obtained investigating *S. cerevisiae* and mammalian Pol I may point to a potential transcription factor-like behavior of the subcomplex. The discussion section has been modified accordingly. Understanding the role of A49/A34.5 subcomplexes in Pol I transcription is important and will remain one focus of our research in the near future.

In addition, the EC structure shows weak density that may be attributed to A12-Ct. A similar position for this domain was observed for *S. cerevisiae* Pol I EC in the presence of GMPCPP (Tafur et al., eLife 2019;8:e43204). A specific comparison between these structures, including additional figure/panels, should be made. Is the position of the A12-Ct in the Tafur paper coherent with the weak density in the EC structure? The authors should also discuss on the functional relevance of this observation and the possible connection with their nucleic acid scaffold, which includes three locked nucleic acid bases at the 3' end of the RNA primer.

The requested figure is now included. We also included the possibility that LNA bases may have an influence on A49/A34.5 dissociation and A12.2C localization in Sp Pol I in the main text. However, this seems unlikely since the same scaffold was previously used in Sc Pol I EC structure determination without presenting such an effect (Neyer et al., Nature 2016).

In general, the manuscript would greatly benefit from a more detailed presentation of results and its discussion, at least for the following sections:

- The conservation of cleft contraction upon nucleic acid binding in the cleft is one of the main observations, but its description and corresponding discussion in the manuscript is succinct. What is the extent of the cleft movement? How does it compare to *S. cerevisiae* Pol I? Are the hinge regions

conserved between the enzymes? Figure S3e should be improved by showing cleft width distances, hinge points, superposition of the pombe and cerevisiae enzymes, etc.

The requested values were previously presented in Supplemental Table 3 and were now included in main Fig. 4. Hinge regions are indicated in the supplemental movies more prominently referenced in the discussion.

- Related to this, helix H0 in *S. cerevisiae* Rpb6 is ordered only in Pol I and not in Pol II or Pol III. This has been connected to increased stalk attachment to the Pol I core and to rigidity of the clamp-shelf module. The authors should describe and discuss whether this is the same in *S. pombe* Pol I. Moreover, the authors should include an alignment of Rpb6 (and also the other 4 subunits shared by the three nuclear RNA polymerases) in Figure S4. Figure S4 should also indicate relevant structural domains discussed in the main text.

We expanded the text section and included alignments of the five common subunits in the supplement for completeness even though the structures were previously solved within Sp Pol II (Spahr et al., PNAS 2010).

- Conformational changes between the free monomeric and dimeric states of the enzyme are also poorly described. In *S. cerevisiae*, further cleft expansion is observed upon dimerization, which associates with binding of the expander loop inside the cleft. Is further cleft expansion and expander loop location conserved in *S. pombe* Pol I? A figure/panel on such conformational changes in *S. pombe* should be added.

As requested by reviewer #1, we added a Figure comparing Sc and Sp Pol I expander/DNA-mimicking loop location. Cleft expansion states were originally included in Sup. Table 3 and are now part of the main text. Indeed, the extreme expansion of the upstream cleft upon Sc Pol I dimerization is not observed in Sp. Likely, this is due to connector association and insertion of the stalk from the neighboring monomer in Sc, which is absent in Sp.

- Finally, the main text contains only 2 figures. Figures S3 and S5 are very relevant and should be moved into the main text.

Originally, we planned on a report-style presentation of our results. Encouraged by reviewer #2 we have expanded the main manuscript to a certain extent.

REVIEWERS' COMMENTS

Reviewer #1 (Remarks to the Author):

These are comments for the revised manuscript. I thank the authors for careful attention made to my comments and am satisfied that they have shown that dimerisation can occur without crosslinking in vitro. Ideally, this would have been followed by testing this in vivo, but as alluded to in their responses to reviewer #2, I agree this is non-trivial and out of scope for this study. The additional edits regarding Rrn3/7 and the expander are also helpful.

I recommend this study for publication.

Reviewer #2 (Remarks to the Author):

The authors have improved the manuscript in several respects to address most of the major criticisms. They have now enriched the description and discussion of their results, especially concerning lack of density for A49/A34.5 and conformational changes between different states, including previously under-described structural elements. Importantly, they now demonstrate that dimers form in vitro in a concentration-dependent manner, further substantiating their putative in vivo relevance. Moreover, the authors initiated in vivo studies to validate their structural results by constructing an A14-delta strain, whose preliminary characterization indicates that dimer formation may not be essential in vivo, which is of interest. Inclusion of a thorough study of this strain would further underscore the relevance of their result. Nevertheless, given the overall significance of the authors' findings, I consider that the revised manuscript is suitable for publication.